# Stateless actor-critic for instance segmentation with high-level priors

## Abstract

Instance segmentation is an important computer vision problem which remains challenging despite impressive recent advances due to deep learning-based methods. Given sufficient training data, fully supervised methods can yield excellent performance, but annotation of ground-truth data remains a major bottleneck, especially for biomedical applications where it has to be performed by domain experts. The amount of labels required can be drastically reduced by using rules derived from prior knowledge to guide the segmentation. However, these rules are in general not differentiable and thus cannot be used with existing methods. Here, we relax this requirement by using stateless actor critic reinforcement learning, which enables non-differentiable rewards. We formulate the instance segmentation problem as graph partitioning and the actor critic predicts the edge weights driven by the rewards, which are based on the conformity of segmented instances to high-level priors on object shape, position or size. The experiments on toy and real datasets demonstrate that we can achieve excellent performance without any direct supervision based only on a rich set of priors.

## 1 Introduction

Instance segmentation is the task of segmenting all objects in an image and assigning each of them a different label. It forms the necessary first step to the analysis of individual objects in a scene and is thus of paramount importance in many practical applications of computer vision. Over the recent years, fully supervised instance segmentation methods have made tremendous progress both in natural image applications and in scientific imaging, achieving excellent segmentations for very difficult tasks [1, 2].

A large corpus of training images is hard to avoid when the segmentation method needs to take into account the full variability of the natural world. However, in many practical segmentation tasks the appearance of the objects can be expected to conform to certain rules which are known *a priori*. Examples include surveillance, industrial quality control and especially medical and biological imaging applications where full exploitation of such prior knowledge is particularly important as the training data is sparse and difficult to acquire: pixelwise annotation of the necessary instance-level groundtruth for a microscopy experiment can take weeks or even months of expert time. The use of shape priors has a strong history in this domain [3, 4], but the most powerful learned shape models still require groundtruth [5] and generic shapes are hard to combine with the CNN losses and other, non-shape, priors. For many high-level priors it has already been demonstrated that integration of the prior directly into the CNN loss can lead to superior segmentations while significantly reducing the necessary amounts of training data [6]. However, the requirement of formulating the prior as a differentiable function poses a severe limitation on the kinds of high-level knowledge that can be exploited with such an approach. The aim of our contribution is to address this limitation and

Submitted to 35th Conference on Neural Information Processing Systems (NeurIPS 2021). Do not distribute.

establish a framework in which a rich set of non-differentiable rules and expectations can be used to steer the network training.

To circumvent the requirement of a differentiable loss function, we turn to the reinforcement learning paradigm, where the rewards can be computed from a non-differentiable cost function. We base our framework on a stateless actor-critic setup [7], providing one of the first practical applications of this important theoretical construct. In more detail, we solve the instance segmentation problem as agglomeration of image superpixels, with the agent predicting the weights of the edges in the superpixel region adjacency graph. Based on the predicted weights, the segmentation is obtained through (non-differentiable) graph partitioning and the segmented objects are then evaluated by the critic, which learns to approximate the rewards based on the object- and image-level reasoning (see Fig. 1).

The main contributions of this work can be summarized as follows: (i) we formulate instance segmentation as a RL problem based on a stateless actor-critic setup, encapsulating the non-differentiable step of instance extraction into the environment and thus achieving end-to-end learning; (ii) we exploit prior knowledge on instance appearance and morphology by tying the rewards to the conformity of the predicted objects to pre-defined rules and learning to approximate the (non-differentiable) reward function with the critic; (iii) we introduce a strategy for spatial decomposition of rewards based on fixed-sized subgraphs to enable localized supervision from combinations of object- and image-level rules. (iv) we demonstrate the feasibility of our approach on synthetic and real images and show an application to an important segmentation task in developmental biology, where our framework delivers an excellent segmentation with no supervision other than high-level rules.

## 2 Related work

Reinforcement learning has so far not found significant adoption in the segmentation domain. The closest to our work are two methods in which RL has been introduced to learn a sequence of segmentation decision steps as a Markov Decision Process. In the actor critic framework of [8], the actor recurrently predicts one instance mask at a time based on the gradient provided by the critic. The training needs fully segmented images as supervision and the overall system, including an LSTM sub-network between the encoder and the decoder, is fairly complex. In [9], the individual decision steps correspond to merges of clusters while their sequence defines a hierarchical agglomeration process on a superpixel graph. The reward function is based on Rand index and thus not differentiable, but the overall framework requires full (super)pixelwise supervision for training.

Reward decomposition was introduced for multi agent RL by [10] where a global reward is decomposed into a per agent reward. [11] proves that a stateless RL setup with decomposed rewards requires far less training samples than a RL setup with a global reward. In [12] reward decomposition is applied both temporally and spatially for zero-shot inference on unseen environments by training on locally selected samples to learn the underlying physics of the environment.

The restriction to differentiable losses is present in all application domains of deep learning. Common ways to address it are usually based on a soft relaxation of the loss that can be differentiated. The relaxation can be designed specifically for the loss, such as, for example, Area-under-Curve [13] for classification or Jaccard Index [14] for semantic segmentation. These approaches are not directly applicable to our use case as we aim to enable the use of a variety of object- and image-level priors which can easily be combined without handcrafting an approximate loss for each case. More generally, but still for a concrete task loss, Direct Loss Minimization has been proposed for CNN training in [15]. For semi-supervised learning of a classification or ranking task, Discriminative Adversarial Networks have been proposed as a means to learn an approximation to the loss [16]. Most generally, Grabocka et al. in [17] propose to train a surrogate neural network which will serve as a smooth approximation of the true loss. In our setup, the critic can informally be viewed as a surrogate network as it learns to approximate the priors through the rewards by Q-learning.

Incorporation of rules and priors is particularly important in biomedical imaging applications, where such knowledge can be exploited to augment or even substitute scarce groundtruth annotations. For example, the shape prior is explicitly encoded in the popular nuclear [18] and cellular [19] segmentation algorithms based on spatial embedding learning. Learned non-linear representations of the shape are used in [5], while in [20] the loss for object boundary prediction is made topology-aware. Domain-specific priors can also be exploited in post-processing by graph partitioning [21]. Interestingly, the energy minimization procedure underlying the graph partitioning can also be incorporated into the learning step [22, 23].

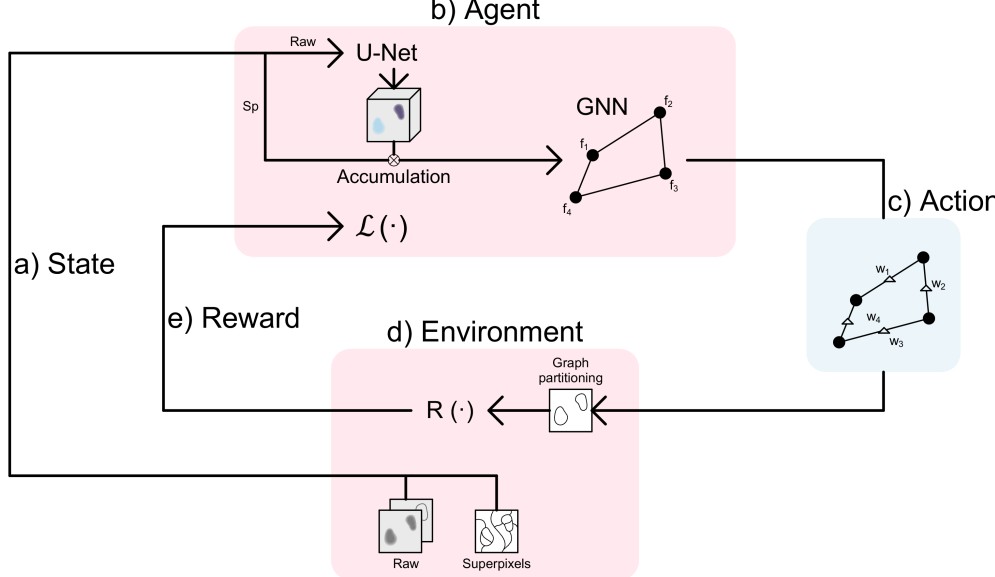

Figure 1: Interaction of the agent with the environment: (a) shows the state which is composed of the raw image and the superpixel over-segmentation; (b) depicts the agent and the superpixel graph, which accumulates the features for nodes of the GNN from pixels which belong to the corresponding superpixels; (c) given the state, the agent performs the actions by predicting edge weights on the superpixel graph; (d) the environment, which includes the graph partitioning built from the weights predicted through agent actions; (e) rewards are obtained by evaluating the segmentation arising from the graph partitioning, based on pre-defined and data dependent rules. The rewards are given back to the agent where they are used for training.

## 3    Methods

The task of instance segmentation can be formalized as transforming an image $x$ into a labeling $y$, where $y$ maps each pixel to a label value. An instance corresponds to the maximal set of pixels with the same label value. Typically, the instance segmentation problem is solved via supervised learning, i.e. using a training set with ground-truth labels $\hat{y}$. Note that $y$ is invariant under the permutation of label values. In general, it is difficult to formulate instance segmentation in a fully differentiable manner. Most approaches first predict a "soft" representation with a CNN, e.g. affinities [1, 24, 25], boundaries [26, 27] or embeddings [28, 29] and apply non-differentiable post-processing, such as agglomeration [27, 30], clustering [31, 32] or partitioning [33], to obtain the instance segmentation. Alternatively, proposal-based methods predict a bounding-box per instance and then predict the instance mask for each bounding-box [34]. Furthermore, the common evaluation metrics for instance segmentation [35, 36] are also not differentiable.

Our main motivation to explore RL for the instance segmentation task is to circumvent the restriction to differentiable losses and - regardless of the loss - to make the whole pipeline differentiable end-to-end even in presence of non-differentiable steps which transform pixelwise CNN predictions into individual instances.

We formulate the instance segmentation problem using a region adjacency graph $G = (V, E)$, where the nodes $V$ correspond to superpixels (homogeneous clusters of pixels) and the edges $E$ connect nodes which belong to spatially adjacent superpixels. Given edge weights $W$, an instance segmentation can be obtained by partitioning the graph, here using an approximate multicut solver [37]. Together, the image data, superpixels, graph and the graph partitioning make up the environment $\mathcal{E}$ of our RL setup. Based on the state $s$ of $\mathcal{E}$, the agent $\mathcal{A}$ predicts actions $a$, which are used to compute the partitioning. The reward $r$ is then computed based on this partitioning. Our agent $\mathcal{A}$ is a stateless actor-critic [38], represented by two graph neural networks (GNN) [39]. The actor predicts the actions $a$ based on the graph and its node features $F$. The node(superpixel) features are computed by pooling together the corrresponding pixel features based on the raw image data.

Here, we make use of two different setups: Method 1, where the per-pixel features are computed based on the image data with the feature extractor being part of the agent $\mathcal{A}$ and Method 2 where the feature extractor is part of the environment $\mathcal{E}$. The feature extractor is trained end-to-end in Method 1, whereas it is fixed and thus needs to be pre-trained in Method 2. We use a U-Net [40] as feature extractor and can use hand-crafted features in addition to the learned features. More details about the pre- training can be found in the Appendix. The agent - environment interaction for Method 1 is depicted in Figure 1. For Method 2 we refer to the Appendix.

Importantly, this setup enables us to use both a non-differentiable instance segmentation step and reward function, by encapsulation of the "pixels to instances" step in the environment and learning a policy based on the rewards with a stateless actor critic.

### 3.1 Stateless Reinforcement Learning Setup

Unlike most RL settings [41], our approach does not require an explicitly time dependent state: the actions returned by the agent correspond to the real-valued edge weights in $[0, 1]$, which are used to compute the graph partitioning. Any state can be reached by a single step from the initial state and there exists no time dependency in the state transition. Unlike [9], we predict all edge values at once which allows us to avoid the iterative strategy of [8] and deliver and evaluate a complete segmentation in every step. We implement a stateless actor critic formulation with episodes of length 1.

To the best of our knowledge, stateless RL was introduced in [7] to study the connection between generative adversarial networks and actor critics and our method is one of the first practical applications of this concept. Here, the agent consists of an actor, which predicts the actions $a$ and a critic, which predicts the action value $Q$ (expected future discounted reward) given the actions. The stateless approach simplifies the action value function: the action value has to estimate the reward for a single step instead of estimating the expected sum of discounted future rewards for many steps. We have explored a multi-step setup as well, but found that it yields inferior results for our application; details can be found in the Appendix. As described in detail in 3.2, we compute localized sub-graph rewards instead of relying on a single global reward.

The actor corresponds to a single GNN, which predicts the mean and variance of a normal distribution for each edge. The actions $a$ are determined by sampling from this distribution and applying a sigmoid to the result to obtain continuous edge weights in the value range $[0, 1]$. The GNN takes the state $s = (G, F)$ as input arguments and its graph convolution for the $i^{th}$ node is defined as in [39]:

$$f_i = \gamma_\pi \left( f_i, \frac{1}{|N(i)|} \sum_{j \in N(i)} \phi_\pi \left( f_i, f_j \right) \right) \tag{1}$$

where $\gamma_\pi$ as well as $\phi_\pi$ are MLPs, $(\cdot, \cdot)$ is the concatenation of vectors and $N(i)$ is the set of neighbors of node $i$. The gradient of the loss for the actor is given by:

$$\nabla_\theta \mathcal{L}_{actor} = \nabla_\theta \frac{1}{|SG|} \sum_{sg \in G} \left[ \alpha \sum_{\hat{a} \in sg} log(\pi^\theta(\hat{a}|s)) - Q_{sg}(s, a) \right] \tag{2}$$

This loss gradient is derived following [38]. We adapt it to the sub-graph reward structure by calculating the joint action probability of the policy $\pi^\theta$ over each sub-graph $sg$ in the set of all sub-graphs $SG$. Using this loss to optimize the policy parameters $\theta$ minimizes the Kullback-Leibler divergence between the Gibbs distribution of action values for each sub-graph $Q_{sg}(s, a)$ and the policy with respect to the parameters $\theta$ of the policy. $\alpha$ is a trainable temperature parameter which is optimized following the method introduced by [38].

The critic predicts the action value $Q_{sg}$ for each sub-graph $sg \in SG$. It consists of a GNN $Q_{sg}(s, a)$ that takes the state $s = (G, F)$ as well as the actions $a$ predicted by the actor as input and predicts a feature vector for each edge. The graph convolution from Equation 2 is slightly modified:

$$f_i = \gamma_Q \left( f_i, \frac{1}{|N(i)|} \sum_{j \in N(i)} \phi_Q \left( f_i, f_j, a_{(i,j)} \right) \right) \tag{3}$$

again $\gamma_Q$ and $\phi_Q$ are MLPs. Based on these edge features $Q_{sg}$ is predicted for each sub-graph via an MLP. Here, we use a set of subgraph sizes (typically, 6, 12, 32, 128) to generate a supervison signal

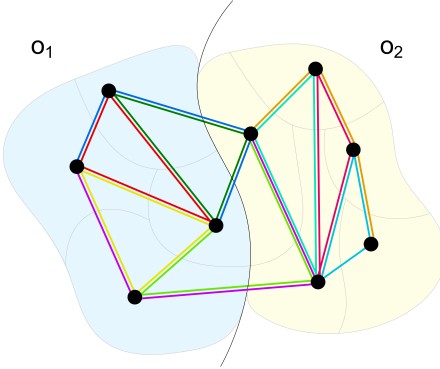

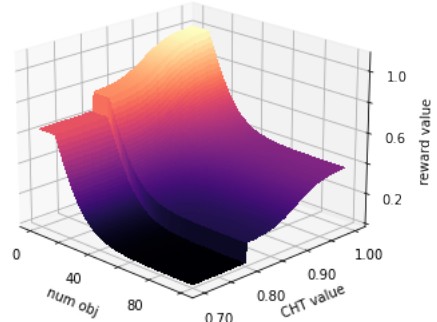

Figure 2: The graph is subdivided into sub-graphs, each sub-graph is highlighted by a different color. All sub-graphs have the same number of edges (here 3). Overall, we use a variety of sizes covering different notions of locality.

Figure 3: An example reward landscape Circle Hough Transform (CHT) rewards. High rewards are given if the overall number of predicted objects is not too high and if the respective object has a large CHT value. We found an exponential gradient of the reward landscape to work best.

for different neighborhood scales. A given MLP is only valid for a fixed graph size, so we employ a different MLP for each size. The loss for the critic is given by:

$$\mathcal{L}_{critic} = \frac{1}{|SG|} \sum_{sg \in G} \frac{1}{2} (Q_{sg}^{\delta}(s, a) - r)^2 \qquad (4)$$

Minimizing this loss with respect to the action value function's parameters $\delta$ minimizes the difference between the expected reward and action values $Q_{sg}^{\delta}(s, a)$.

## 3.2 Localized Supervision Signals

The RL paradigm is to provide a global reward for a given state transition [41]. However, we find that for our application it is possible and desirable to instead provide several more localized rewards per state transition: Given a large action space with a policy represented by a complex multivariate probability distribution, it is beneficial to learn from rewards for the specific actions rather than from a scalar global reward for the union of all actions. Of course then requirement arises that the union of local rewards must resemble to the global reward. E.g. the optimal policy is the same for local as for the global reward.

Our actor critic setup (Section 3.1) expects rewards per sub-graph. A good set of sub-graphs should fulfill the following requirements: each sub-graph should be connected so that the information presented to the MLP computing the action value for this sub-graph is correlated. The size of the sub-graphs, given by the number of edges, should be a parameter and all sub-graphs should be extracted with exactly that size to serve as valid input for one of the MLPs. The union of all sub-graphs should cover the complete graph so that each edge contributes to at least one action value $Q_{sg}$. The sub-graphs should overlap to provide a smooth sum of action values. We introduce Algorithm 1 to extract such a set of sub-graphs (see Appendix). Figure 2 shows the sub-graphs for a small example graph.

While some of the rewards used in our experiments can be directly defined for the sub-graphs, most are instead defined per object (see Appendix for details on reward design). We use the following general procedure to map object-level rewards to sub-graphs: first assign to each superpixel the reward of its corresponding object, then determine the reward per edge as the maximum value of its two incident superpixels' rewards and average the edge rewards to obtain the reward per sub-graph. Here, we use the maximum because high object scores indicate that all actions contributing to the respective object should get a high reward. However, for low object scores it is not possible to localize the specific action responsible for the low score. Hence, by taking the maximum we assign the higher score to edges whose incident superpixels belong to different objects, because they probably correspond to a correct split. Note that the uncertainty in the assignment of low rewards can lead to a noisy reward signal, but the averaging of the edge rewards over the sub-graphs and the overlaps

between the sub-graphs smooth and partially denoise the rewards. We have also explored a different actor critic setup that can use object level rewards directly, eliminating the need for the sub-graph extraction and mapping. However, this approach yields inferior results, see the Appendix for details.

## 4 Experiments

The agent of our setup acts on the superpixel graph and thus depends on the features assigned to the nodes of the graph. We introduced two variants of our algorithm: in the base variant (Method 1) we start from random features and make them part of the agent, allowing them to change through back-propagation (Fig. 1). In contrast, Method 2 acts on predefined features which are provided as part of the environment and are computed before training, e.g. through unsupervised clustering. A very accurate clustering in the features produces an easy problem for the agent to solve where even a global reward for all actions might be sufficient. However, in a real-world setting with no supervision, the noisier the features become the more local the reward has to be. We evaluate Method 2 on synthetic data where self-supervised pretraining can deliver noisy, but meaningful node features. Our full setup with Method 1 is evaluated on a dataset from a light microscopy experiment, where highly regular object shapes are to be expected, but no good feature pre-training is possible.

To transform the edge weight predictions of the agent into an instance segmentation we use the Multicut [42] algorithm. Here, other options are also possible such as hierarchical clustering used in [9], but we choose the Multicut for its global optimality property. Hyperparameters of the pipeline were found by cross-validation (see Appendix).

### 4.1 Synthetic dataset: circles on structured ground

To evaluate the feasibility of our approach, we create a synthetic dataset with prominent structured background. Our aim is to segment irregular disks on such background using only rule-based supervision. We generate the superpixels by the mutex watershed algorithm [25] which we run on the Gaussian gradient image. The node features of the superpixel graph were computed through self-supervised pretraining with contrastive loss as described in Appendix and fixed as part of the environment.

As we aim to segment disks, we compute the circularity of the segmented objects for the rewards using the Circle Hough Transform [43]. This object-level reward is combined with the global rough estimate of the number of objects in the image to create the reward surface depicted in Fig. 3. The reward for the number of objects provides useful gradient during early training stages: for example, when too few potential objects are found in the prediction, a low reward can be given to what is thought to be the background object. On the other hand, if too many potential objects are found, a low reward can be given to all the foreground objects with a low CHT value.

In more detail, the object rewards $r_{fg}$ are composed as follows. We define a threshold $\gamma$ on the CHT value ($\gamma = 0.8$ in the reward surface shown in Fig. 3). Let $c \in [0, 1]$ be the CHT value corresponding to the object and let $k$ be the total number of objects that we expect and $n$ be the number of predicted objects. Then

$$r_{local} = \begin{cases} \sigma\left(\left(\frac{c-\gamma}{1-\gamma} - 0.5\right)6\right)0.4, & \text{if } c \geq \gamma \\ 0, & \text{otw} \end{cases} \tag{5}$$

$$r_{global} = \begin{cases} r_{exp}\left(\frac{k}{n}\right), & \text{if } n \geq k \\ 0.6, & \text{otw} \end{cases} \tag{6}$$

$$r_{fg} = r_{local} + r_{global} \tag{7}$$

Here $\sigma(\cdot)$ is the sigmoid function. The input to the sigmoid function is normalized to the interval $[-3, 3]$ which was empirically found to be a good range. The rewards are always in $[0, 1]$ here this is split up into $[0, 0.5]$ for the local reward as well as for the global reward.

For the largest predicted object we strongly suspect the background object. For this object background rewards $r_{bg}$ are calculated by

$$r_{bg} = \begin{cases} r_{exp}\left(\frac{n}{k}\right), & \text{if } n \leq k \\ 1, & \text{otw} \end{cases} \tag{8}$$

Note that this rewards have a large globally calculated part which makes this setup not fit for Method 1. It needs some feature representation that already gives a good idea for the clustering. The only

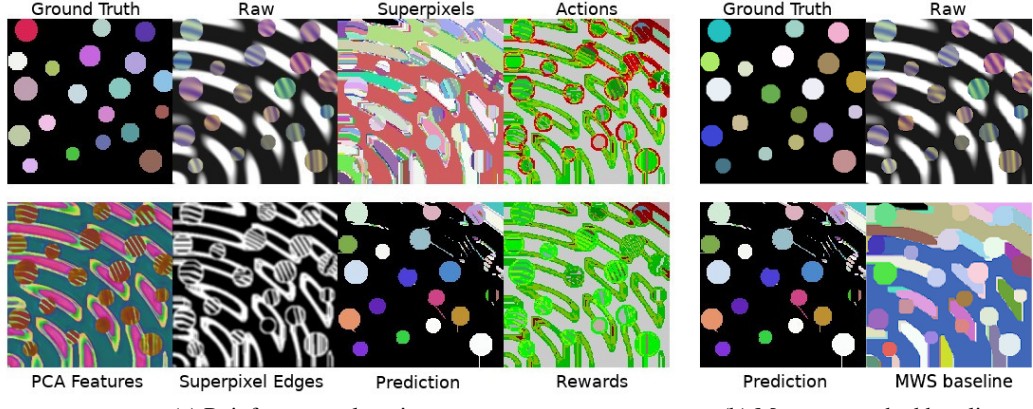

| (a) Reinforcement learning output. | (b) Mutex watershed baseline. |

Figure 4: The "Circles" dataset. Top left to right: ground truth segmentation, raw data, superpixel over-segmentation and a visualizataion for the actions on every edge, where a merge action is displayed in green and a split action in red. Bottom left to right: the pre-trained pixel embeddings projected to their first 3 PCA components shown as RGB, an edge image of the superpixels, the segmentation resulting from the graph agglomeration on the predicted edge weights and a visualization of the rewards based on the CHT, where light green shows high rewards and dark red low rewards.

useful local information in the reward is the CHT value. Therefore, if the features have a fairly distinct structure for circles, the agent should be able to find and to correctly cluster them.

Fig. 4 shows the output of all algorithm components on a sample image. For comparison, we also computed mutex watershed [25] predictions. Texture within objects and structured background are inherently difficult for region-growing algorithms, but our approach can exploit higher-level reasoning along with low-level information and achieve a good segmentation.

## 4.2 Real dataset: light microscopy imaging

Biomedical applications often require segmentation of objects of known morphology which are positioned in regular patterns, while extensive prior knowledge is available on variability of both under normal experimental conditions [44]. Such data presents the best use case for our algorithm as the reward function can leverage the known characteristics of individual object shape and texture and the overall similarity of the objects.

The dataset used for this experiment contains 317 2D images extracted from a video of a developing fruitfly embryo acquired with a light-sheet microscope [45] (Fig. 5). The image shows boundaries (plasma membranes) of the embryo cells. Across the dataset, 10 images were fully segmented by an expert, we use those for validation.

Fruitfly embryo is a well-studied system for which we can exploit the prior knowledge on the expected cell shape and the radial pattern of cells. Furthermore, as the analysis of cell shape dynamics is a paramount part of many biological experiments, multiple pre-trained networks are available for the cell segmentation task [18, 19, 46, 47]. Due to the differences in sample preparation and image acquisition settings, none of these would work out-of-the-box for our data. However, the CNNs in [47] which are trained to predict boundaries in confocal microscope images of plant tissue, can serve as a strong edge detector to create superpixels in our images. The superpixels are obtained using the seeded watershed algorithm on seeds at the local minima of the predicted edge map.

The rewards for this experiment are designed as follows: we set a high reward for merging the superpixels which are certain to lie in the background (close to the image boundary or the image center). For the background edges near the foreground area we modulate the reward by the circularity of the overall foreground contour. Finally, for the edges which are likely to be in the foreground we compute object-level rewards by fitting a rotated bounding box to each object and comparing its side lengths as well as its orientation to predefined template values. We do not perform semantic segmentation to define precise foreground/background boundaries, but instead use a soft weighting scheme with Gaussian weights to combine object and background rewards based on on the prior

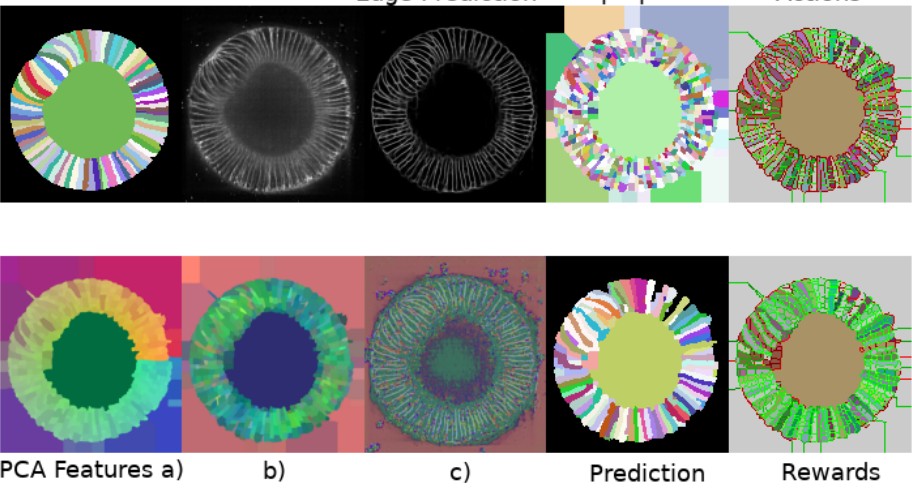

Figure 5: Microscopy dataset experiment. Top left to right: ground truth segmentation; raw data; edge map; superpixel over-segmentation; visualization for the actions on every edge, where a merge action is displayed in green and a split action in red. Bottom left to right: a) handcrafted features; b) learned features accumulated on superpixels; c) learned features projected to their first 3 PCA components shown as RGB; the segmentation resulting from the Multicut on the predicted edge weights; visualization of the rewards, where light green shows high rewards and dark red low rewards.

271 knowledge of the embryo width. An image of the weights for different locations in the image can be
272 found in the appendix.

273 More formally the edge rewards $r_{edge}$ are calculated as follows. For each edge, we define the distance
274 $h$ between the edge and the center of the image as the average distance of the incident objects' center
275 of mass and the center $c$ of the image. $j$ is the approximate radius of the circle that lies within the
276 foreground and $m$ is the maximal distance between $c$ and the image boarder. Let further $\mathcal{K}(\cdot)$ be the
277 Gaussian kernel function. Then $r_{edge}$ yields

$$r_{bg} = \begin{cases} \mathcal{K}\left(\frac{||h-c||}{\gamma}\right)(1-a), & \text{if } h \leq j \\ \mathcal{K}\left(\frac{||m-h||}{\eta}\right)(1-a), & \text{otw} \end{cases} \tag{9}$$

$$r_{fg} = \mathcal{K}\left(\frac{||h-j||}{\delta}\right) max(r_{o1}, r_{o2}) \tag{10}$$

$$r_{edge} = r_{fg} + r_{bg} \tag{11}$$

278 Here $\gamma, \eta, \delta$ are normalization constants. Equation 9 first determines the background probability for
279 an edge by the kernel values. $1-a$ constitutes a reward that directly favors merges which is scaled
280 by the background probability. For each edge, $r_{o1}$ and $r_{o2}$ are the rewards corresponding to the two
281 objects connected to that edge. The object rewards are given by fitting a rotated bounding box to the
282 object and then compare rotation and dimensions to template values.

283 Note that in this experiment no self-supervised pretraining is used for the node features in the
284 agent's GNNs. Unlike the "Circles" dataset, all objects in these images have very similar intensity
285 distributions and can only be separated through the detection of boundaries between them. Instead
286 of the pretraining, we experiment with using a few hand-crafted features like the polar coordinate
287 of the node's respective superpixel's center of mass with respect to the coordinate system sitting
288 at the center of the image as well as the superpixel's mass, and with learning other features by
289 back-propagation from the agent. The handcrafted features are normalized, concatenated to the
290 learned features and used as input to the GNN. The projection of the first 3 PCA components of these
291 features into RGB space is shown in Fig. 5 respectively for learned feature maps, their projection
292 to node features through the accumulation procedure and finally the concatenation of those and the

Table 1: Quantitative evaluation on the microscopy dataset. Note that the projection of superpixels to the ground truth (sp gt) sets an upper (lower for VI) bound for our method. We use Symmetric Best Dice as well as the Variation of Information metric to compare all results on the validation set.

| Method | SBD | VI merge | VI split |
|---|---|---|---|
| sp gt | $0.656 \pm 0.019$ | $0.672 \pm 0.061$ | $0.594 \pm 0.028$ |
| ours + augmentation noise | $\underline{0.508 \pm 0.031}$ | $1.233 \pm 0.156$ | $1.060 \pm 0.258$ |
| ours | $0.482 \pm 0.020$ | $\underline{0.839 \pm 0.118}$ | $1.374 \pm 0.357$ |
| ours without edges | $0.446 \pm 0.041$ | $0.953 \pm 0.212$ | $0.994 \pm 0.200$ |
| ours only handcrafted | $0.408 \pm 0.087$ | $0.987 \pm 0.101$ | $1.536 \pm 0.410$ |
| edge + mc [47] | $0.283 \pm 0.023$ | $3.019 \pm 0.040$ | $\underline{0.342 \pm 0.045}$ |
| contrastive [28] | $0.215 \pm 0.009$ | $1.155 \pm 0.037$ | $3.285 \pm 0.084$ |
| contrastive + edge [28] | $0.248 \pm 0.014$ | $1.229 \pm 0.048$ | $3.336 \pm 0.073$ |

handcrafted features. Note that the learned features converge to a representation which resembles a semantic segmentation of boundaries in the image.

We train the complete setup for Method 1 end-to-end on a Nvidia GeForce RTX 3090 GPU for 4 days. For comparison we keep the model which achieved the highest reward on the test set. This makes training as well as the validation independent from ground truth annotations. The evolution of the rewards on the validation set for different random seeds is shown in the Appendix. All of the conducted trainings show a stride for high rewards regardless of different random seeding.

For the validation scores we use the variation of information (VI) for both input combinations (merge and split) and the Symmetric Best Dice score. To show the influence of the imperfect superpixels on the final clustering, we project the superpixels to their respective ground truth clustering ("sp gt" in Table 1) which sets an upper (lower in case of VI) bound for our method. In this study we use several versions of our approach. In Table 1 (ours) refers to method 1 as described in section 4.2, (ours + augmentation noise) is the same method but add some noise to the input data during training, (ours without edges) is our method but without the additional edge prediction as an input and (ours only handcrafted) is our method where we only use the handcrafted features as described in section 4.2. We find that learned features significantly contribute to the performance of our method.
We compare to the following baseline approaches: *edge + mc*, which solves the Multicut graph partitioning based on edge weights derived from boundary predictions used for superpixel creation, *contrastive*, which predicts a pixel-wise embedding space that is clustered into instances using k-means and for which the embeddings are trained using the discriminative loss function of [28] on the ovules dataset from [47] and *contrastive + edge*, which is similar to *contrastive*, but receives the [47] boundary predictions as additional input channel.

## 5   Discussion

We introduced an end-to-end instance segmentation algorithm which can exploit non-differentiable loss functions and high-level prior information. Our RL approach is based on stateless actor-critic and predicts the full segmentation at every step, allowing us to assign rewards to all objects and reach stable convergence. The segmentation problem is formulated as graph partitioning; we design a reward decomposition algorithm which maps object- and image-level rewards to sub-graphs for localized supervision.

We performed proof-of-concept experiments to demonstrate the feasibility of our approach on synthetic and real data and showed in particular that our setup can segment microscopy images with no direct supervision other than high-level reasoning. In the future, we plan to explore other problems and reward functions as well as a semi-supervised setup (briefly introduced in Appendix) where we think our approach can be very beneficial. Furthermore, even in case of full supervision with ample groundtruth, our RL-based formulation enables end-to-end instance segmentation with direct object-level reasoning, which will allow for post-processing-aware training of the CNN which predicts object boundaries or embeddings.

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
