## A Appendix

### A.1 Method 2

In Method 2 the node features are not learned end-to-end by the agent, but instead generated as part of the environment; see also Figure 6. To this end, we employ self-supervised learning to generate a superpixel embedding, as explained in Section A.2. In case of a weak reward function, we find Method 2 preferable over Method 1 which jointly learns the node features. To generate a well performing agent with our method, the reward signal needs to be somewhat close to the "true" metric when evaluating a proposed segmentation. If it is close enough, e.g. by direct supervision or fairly specific priors such as the ones we use for the fruitfly dataset, we can use a weak state representation and learn how to extract the node features jointly (Method 1). Conversely, if the reward does not fit the "true" metric well, it is beneficial to remove the complexity of joint learning and provide a more informative state representation (Method 2).

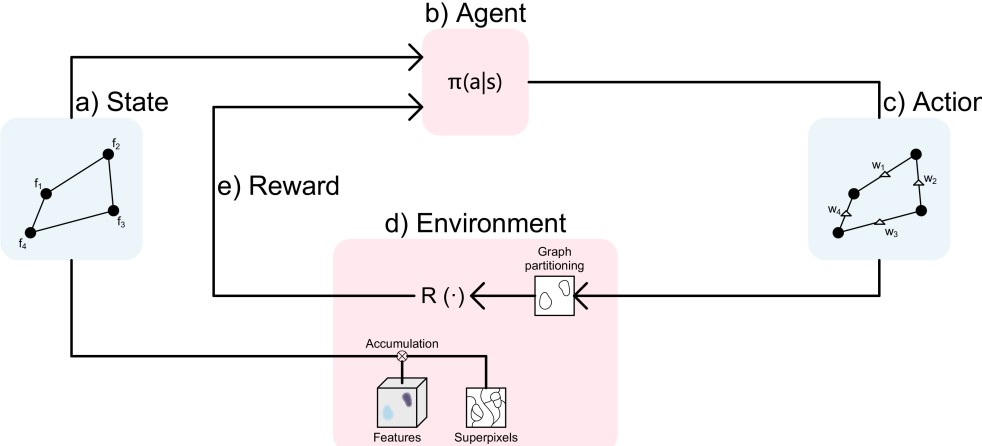

Figure 6: Interaction of the agent with the environment for Method 2: (a) shows the state which is composed of the superpixel graph where each node comes with a feature vector encoding features for the respective superpixel. This features are obtained by accumulation of pre-trained pixel-level features per superpixel; Given the state, the agent (b) predicts edge weights on the graph; These edge weights serve as the actions (c); the actions are processed by a graph partitioning algorithm which is part of the environment (d); (e) shows the rewards, based on the resulting segmentation. Rewards are obtained using a set of predefined and data dependent rules. The rewards are given back to the agent and the episode terminates.

### A.2 Self-supervised pretraining

For self-supervised pre-training, we use a method based on the contrastive loss formulation of [49]. Consider a graph $G = (V, E)$, where the nodes in $V = \{1, 2, ..., n\}$ correspond to the individual superpixels and the edges in $E = \{(i, j) | i \neq j \text{ and } i, j \in V\}$ connect nodes with adjacent superpixels. In addition, consider edge weights $W \in \mathbb{R}^{|E|}$ associated with every edge. Here, we infer the weights from pixel-wise boundary probability predictions and normalize the weights such that $\sum_{w \in W} w = 1$ holds. We train a 2D U-Net to predict embeddings for each node in $V$ by pulling together pixel embeddings that belong to the same superpixel and pushing apart pixel embeddings for *adjacent* superpixels. The intensity of the push force is scaled by the weight of the respective edge. With pixel embeddings $x_n$ and node embeddings $f_i = \frac{1}{m_i} \sum_{k \in s_i} x_k$, where $m_i$ is the mass of the superpixel for node $i$ and $s_i$ is the set of indices for all pixels of the corresponding superpixel, and in accordance

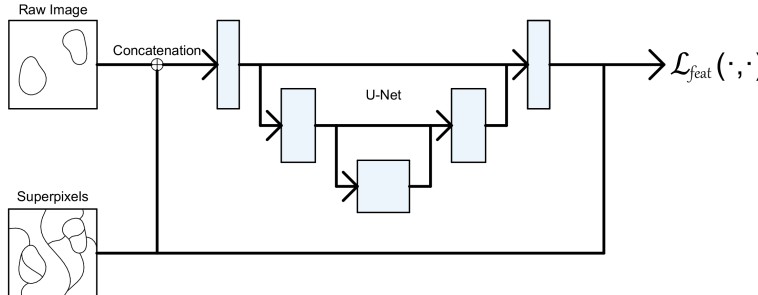

Figure 7: Training setup of the feature extractor. The input is a concatenation of the raw data and a smoothed edge map of the superpixels. The superpixel over-segmentation is used in the loss again as the supervision for learning the embedding space.

with [49] we formulate the loss as

$$\mathcal{L}_{var} = \frac{1}{|N|} \sum_{i=1}^{|N|} \frac{1}{m_i} \sum_{n=1}^{m_i} [d(f_i, x_n) - \delta_v]_+^2 \tag{12}$$

$$\mathcal{L}_{dist} = \sum_{(i,j) \in E}^{|E|} w_{(i,j)} [2\delta_d - d(f_i, f_j)]_+^2 \tag{13}$$

$$\mathcal{L}_{feat} = \mathcal{L}_{var} + \mathcal{L}_{dist} \tag{14}$$

Here $[\cdot]_+$ refers to selecting the max value from the argument and $0$. The forces are hinged by the distance limits $\delta_{var}$ and $\delta_{dist}$. $d(\cdot)$ refers to the distance function in the embedding space. Since the feature extractor is trained self-supervised, we give it a smooth edge map of the superpixels as well as the raw data as an input, see Figure 7.
The training of the feature extractor happens prior to training the agent, see also Section A.1.

**A.3 Reward Generation**

We seek to express the rewards based on prior rules derived from topology, shape, texture, etc. Rules are typically formulated per-object, Section A.8 describes the object-to-sub-graph reward mapping. The reward function is part of the environment and the critic learns to approximate it via Q-learning, enabling the use of non-differentiable functions.

This approach can also be extended to semantic instance segmentation where in addition to the instance labeling a semantic label is to be predicted. To this end, each predicted object is softly assigned to one of the possible classes and the reward is generated specifically for the predicted class. We make use of this extension by separating the objects into a foreground and background class in our experiments.

In addition to the sub-graph rewards our approach can also be extended to global rewards by global pooling of the output of the critic GNN and adding the squared difference of global action value and reward to Equation 4. Alternatively, the global reward can be distributed onto the sub-graph rewards via a weighted sum of sub-graph reward and global reward. In the second approach a different global reward can be specified per class in the case of the semantic instance segmentation formulation. We make use of the per class global reward to encode a reward for the correct number of predicted objects in our experiments.

The biggest challenge in designing the reward function is to avoid local optima. Since the reward is derived from each predicted object, we define the reward by extracting shape features, position, orientation and size of objects and compare them with our expectation of the true object's features. This similarity score should be monotonically increasing as the objects fit our expectation better. All used similarity functions are to a certain extend linear, however an exponential reward function can speed up learning significantly. Consider an object level reward $r \in [0, 1]$, which is linear. We

Figure 8: Object level rewards. We accumulate edge rewards over each object where we consider all edges that have at least one node within the respective object. E.g. for $o_1$ we consider all edges that are covered by the light blue object as well as all the red "unmerge" edges.

calculate the exponential reward by

$$r_{exp}(r) = \frac{exp(r\theta)}{exp(\theta)} \tag{15}$$

where the factor $\theta$ determines the range of the gradient in the output. We also find that it is better to compute the reward as a "distance function" of all relevant features rather than decomposing it into the features and simply summing up the corresponding rewards. In our experiments the latter approach behaved quite unpredictably and often generated local optima which the agent could not escape.

## A.4 Object level rewards

We have tested generating the rewards based directly on the object scores instead of using the subgraph decomposition described in Section A.8. Since rewards are mainly derived from the features of the predicted objects it seems reasonable to formulate the supervision signal based directly on those objects. To this end we calculate a scalar reward per object as sketched in Figure 8. In this setting, the agent needs the information about the predicted objects when it learns from its own actions, which is in contradiction to the usual RL paradigm since the critic needs the predicted objects to predict the action values. However, the critic is not used during exploration where the objects for the explored actions are already available and can be used to predict action values. In this case, the critic uses a second GNN to predict the per-object action values. It is applied to an object's subgraph, which is composed of all edges that have at least one node in common with the respective object. The graph convolutions are followed by a global pooling operation which yields the scalar action value. This GNN replaces the MLPs used in the case of the reward subgraph decomposition. After extensive testing, we found that this approach is always inferior to the subgraph decomposition.

## A.5 Impact of different feature space capacities

In Table 2 we compare the performance of our method, using different numbers of dimensions in the space of the learned node features (the number of channels in the output of the feature extractor U-Net). We find that the reduced capacity of small feature spaces helps the agent perform better. Here we train and evaluate on the fruit fly embryo dataset.

Table 2: Quantitative evaluation of our method using different feature space dimensionality. We use Symmetric Best Dice as well as the Variation of Information metric to compare all results on the validation set.

| n feature channels | SBD | VI merge | VI split |
|---|---|---|---|
| 4 | $0.518 \pm 0.018$ | $0.889 \pm 0.081$ | $0.823 \pm 0.135$ |
| 12 | $0.493 \pm 0.026$ | $0.821 \pm 0.158$ | $1.241 \pm 0.220$ |
| 16 | $0.482 \pm 0.020$ | $0.838 \pm 0.118$ | $1.374 \pm 0.357$ |

## A.6   Random seed evaluation

Figure 9 shows the training evolution of the average subgraph reward from different random seeds. The model performance depends on the chosen seed and for the final comparisons we select the runs based on the best score. The seed is generated randomly on each run.

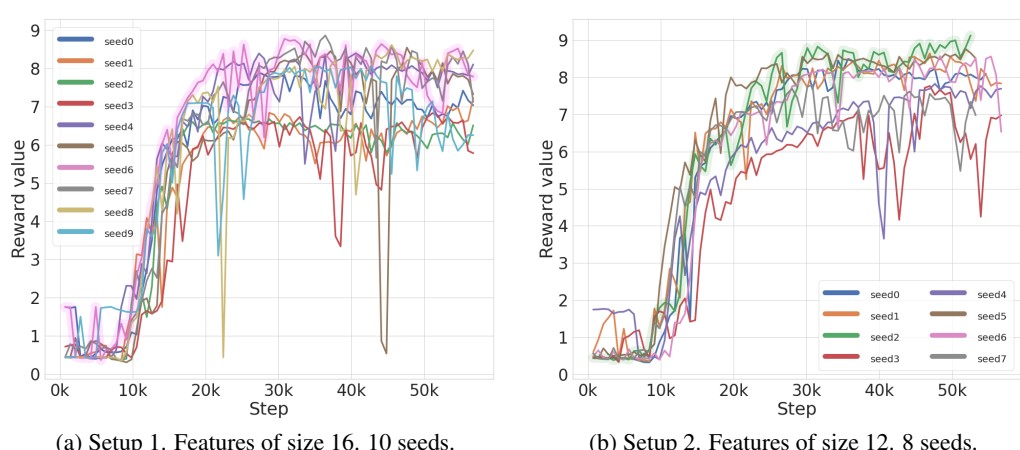

(a) Setup 1. Features of size 16. 10 seeds.          (b) Setup 2. Features of size 12. 8 seeds.

Figure 9: Running the same setup from different random seeds reveals a stable stride for high rewards. We select the model for comparison based on the best achieved reward (magenta line in Fig. 9a and green line in Fig. 9b) which makes the training/validation process completely independent on any ground truth annotations.

## A.7   Gaussian weighting scheme

Figure 10 shows the Gaussian weighting scheme which was used to generate the rewards for the fruitfly embryo data. It can be seen as a very approximate semantic segmentation and serves the purpose of generating a reward maximum at the very approximate segmentation without using it.

## A.8   Randomly generated subgraphs

We select subgraphs using Algorithm 1. Subgraphs are selected randomly starting from random nodes and continuously adding edges to the subgraph until the desired size is reached. The size of the subgraph is defined by the number of edges in the graph. Algorithm 1 selects edges such that the subgraphs are connected and such that their density is high (low number of nodes in the subgraph).

## A.9   Multistep Reinforcement Learning

We tested several methods that use multiple steps within one episode. In this formulation we predict the changes starting from an initial state rather than predicting absolute values for the edge weights. For example, we can start from a state defined by edge weights derived from a boundary map. Given that this state should be somewhat close to the desired state we expect that a few small steps within

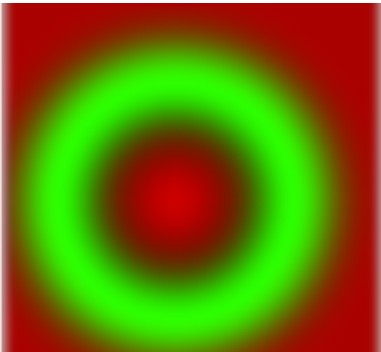 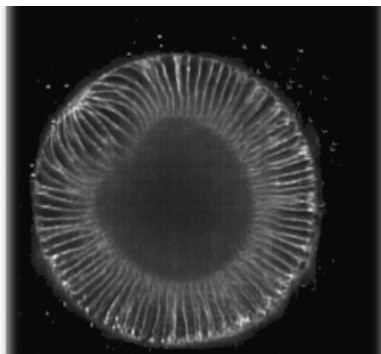

Figure 10: Weighting scheme for object rewards and merge affinity rewards, roughly encoding foreground location. Left: weights for object rewards in green and for merge affinity rewards in red, both are Gaussian and concentric. Right: an example of an underlying real image.

---

**Algorithm 1:** Dense subgraphs in a rag

**Data:** $G = (V, E), l$
**Result:** subgraphs by sets of $l$ edges

1  Initialization:$SG = \emptyset$;
2  **while** $E \backslash SG \neq \emptyset$ **do**
3      pq = PriorityQueue;
4      prio = 0;
5      n_draws = 0;
6      $sg = \emptyset$;
7      $sg_{vtx} = \emptyset$;
8      $i, j = (ij)$ s.t. $(ij) \in E \backslash SG$;
9      pq.push($i$, prio);
10     pq.push($j$, prio);
11     $sg = sg \cup (ij)$;
12     $sg_{vtx} = sg_{vtx} \cup i$;
13     $sg_{vtx} = sg_{vtx} \cup j$;
14     **while** $|sg| < l$ **do**
15         $n$, n_prio = pq.pop();
16         n_draws ++;
17         $adj = \{(nj)|\exists (nj) \in E \text{ and } \exists j \in sg_{vtx}\}$;
18         **forall** $(nj) \in adj$ **do**
19             $sg = sg \cup (nj)$;
20             n_draws = 0;
21         **if** $|adj| < deg(n)$ **then**
22             n_prio -= $(|adj| - 1)$;
23             pq.push($n$, n_prio);
24         **if** $pq.size() \leq n\_draws$ & $\exists j|(nj) \in E, j \notin sg_{vtx}$ **then**
25             $j \in \{j|(nj) \in E, j \notin sg_{vtx}\}$;
26             prio ++;
27             pq.push($j$, prio);
28             $sg = sg \cup (nj))$;
29             $sg_{vtx} = sg_{vtx} \cup j$;
30     $SG = sg \cup SG$
31 **return** $SG$

one episode should be sufficient. In our experiments, we have typically used three steps per episode and used actions that can change the weight per edge by the values in $[-0.1, 0.1]$.

This approach generates an action space that is exponentially larger than in the stateless formulation. A priori this setup might still be more stable because it is not possible to diverge from a given solution so fast due to the incremental changes per step. Also consider the linearity of the paths to the optimal state, which can be generated by giving a high quality reward at every step and not only for the final segmentation. Take for example an initial edge with weight 0.3 and its respective ground truth edge with value 0. We can give a reward that mirrors the correct confidence of the action, which in this case would be the negative action value $r = -a$. This allows us to set the discount factor of the RL setup to 0, because the path to the correct edge weight will be linear and the correct direction will be encoded in the reward at every step. Therefore the rewards for the following steps are not needed. Setting the discount to 0 generates a problem of equal size as the single step RL method, with the disadvantage that the ground truth direction of the path for each edge must be known. Therefore this setup is limited to fully supervised rewards only.

Another possible setup is to give a constant reward at each non terminal step and an unsupervised one at the terminal step with a discount factor $\gamma > 0$. We tested this setup extensively against the stateless setup and found that it was not competitive.

## A.10 Hyper-parameters and network details

For U-Net, we use the standard implementation with features maps size of 32, 64, and 128. The backbone for both actor and critic are GNNs. In addition, the critic employs different MLPs, one for each subgraph size. Most of the hyperparameters for actor, critic and U-Net are chosen empirically and given in the configuration files in the main repository.

The source code with brief instructions is posted on Anonymized GitHub [1]. The dataset is temporarily available on Google Cloud [2].

## A.11 Direct supervision

The aim of this project has been to find a way to train a segmentation algorithm from not-necessarily-differentiable rules, priors and expectations for the segmented objects. Looking ahead to the next stages, we have also qualitatively evaluated the behavior of our setup in case direct supervision such as fully segmented images is available. We tried both full supervision (Fig. 11) and mixed supervision, using one fully segmented image and also the prior rules (Fig. 12). Under full supervision with a set of ground-truth edge weights, we compute the the Dice Score [50] of the predicted edge weights $a$ and the ground-truth $\hat{a}$ for each sub-graph and use it as reward. We find this function to be robust by class imbalance present in our setup. In both cases, the agent learns to segment the circles correctly, demonstrating fast and robust convergence. Note, how learned pixel features converge to a state which strongly resembles a semantic segmentation of the image.

---

[1] https://anonymous.4open.science/r/nips_paper_rlforseg/

[2] https://drive.google.com/file/d/1mHzOuIcLmZP3o4Z_F53bIHzAe6le9vSS

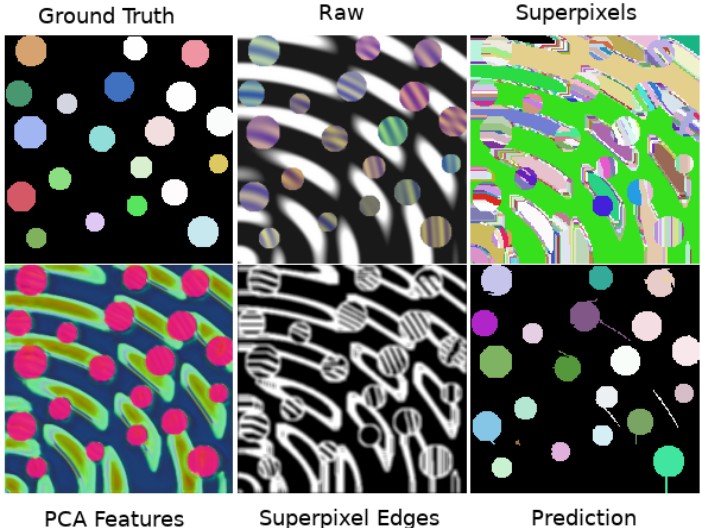

Figure 11: An example of a fully supervised prediction on the "Circles" dataset. This was obtained with use of the Dice score over subgraphs as a fully supervised reward using Method 1. We initialized the the feature extractor U-Net with the pretrained embeddings from Subsection 4.1. It is interesting to see how the features for the circles are emphasized a lot more after training.

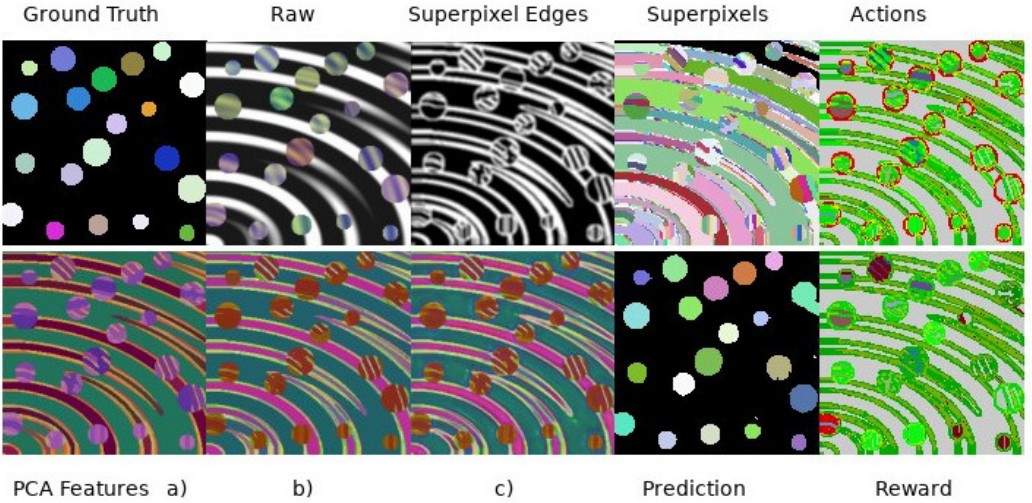

Figure 12: An example for prediction with mixed supervision on the "Circles" dataset. The reward was defined as follows: we use for all but for one image the unsupervised CHT reward and for one image we make use of ground truth and the Dice score as the reward. We find that this mixed reward setting leads to improved performance compared to the unsupervised CHT reward.