# OpenReview forum: "Reinforcement learning for instance segmentation with high-level priors"
_NeurIPS.cc/2021/Conference — NeurIPS 2021 Submitted_

### Official Review · Reviewer_mhie · 2021-07-15

**Rating:** 7
**Confidence:** 4

**Summary:**

This paper presents a strategy for object instance segmentation from images using training data that has no manual annotations. Instead of manual annotations, as is required for many state-of-the-art methods, this paper proposes to use high-level priors represented as rules that can be evaluated in the produced segmentation. These rules define what an acceptable segmentation is, and are used to create a reward signal for a reinforcement learning agent. The framework formulates an environment, an agent, an action space and a reward system in a stateless system to guide segmentation. All the components of the system are well designed and well thought, and this approach may have wide applications in bioimage segmentation problems.

**Ethical Concerns:**

No concerns.

**Limitations And Societal Impact:**

No comments about limitations. Positive societal impacts expected, as less annotations would be required from experts to obtain accurate models.

**Main Review:**

* The stateless actor-critic model has connections with the dynamics of Generative Adversarial Networks, as the authors note. The fact that there is no state also begs the question of what are the connections with the REINFORCE rule for training models when the loss function is non-differentiable, and how that approach could simplify or enhance the proposed formulation.
* The experimental design is interesting and overall complete. However, the performance metrics used for evaluation are not in line with research in the instance segmentation field, where precision and recall are used in several different ways. For example, computing precision or F1 score at specific intersection over union thresholds would be more informative to understand how many objects / regions are correctly segmented and to what level of overlap with ground truth.
* There are a few datasets studied in the instance segmentation literature that could have been used as benchmarks in this evaluation. At least one dataset used for nucleus segmentation of cells, for instance, would add a lot of value to the paper for two reasons: first, it connects with previous literature in the topic, second, it allows to assess how far a model trained without ground truth data would go for segmenting these objects. It is not expected that the proposed model should be better than fully supervised approaches, but being the first in the unsupervised regime would help understand how far we are from a competitive model while not requiring ground truth data.
* The paper is overall well written and clear.

**Time Spent Reviewing:**

3

---

> ### Author Response · Authors · 2021-08-10
> **Connection to REINFORCE; Metrics; Additional Experiments**
>
> We thank the reviewer for the accurate summary and assessment of our manuscript. Here is our point-by-point response:
> 1. As the reviewer correctly points out, we make use of a stateless actor critic similar to [7], where it is used to draw connections between actor critic methods and generative adversarial networks. The question whether we could also use REINFORCE instead of the more complex actor critic is really interesting; we have not considered this option so far because we drew inspiration from [7]. In fact stateless REINFORCE can be connected in a similar way to binary cross entropy as A2C is connected to GANs in [7]. The REINFORCE rule $\sum_t \nabla ~ \textrm{log}(pi(a_t|s_t)) * G_t$ can, for stateless RL, be reformulated to $\nabla ~ \textrm{log}(pi(a|s)) * G$ where $G=R$ is the expected reward. If we see the reward as ground-truth replacement by setting it to 1 for success and to 0 for failure then this loss would resolve to the binary cross entropy loss.
> In practice, the reward should be more expressive and take values in the whole range of [0, 1]. We tested a single network setting like Q-learning and found that our noisy reward generates high fluctuations in the gradient updates and we could not achieve any convergence to meaningful results. As REINFORCE is a single network setting, we expect to experience similar problems as well, but it would be worth a try. In our setup the critic learns to approximate the rewards, which are derived from the instance segmentation scored by our prior rules. Both the instance segmentation algorithm, which operates on the outputs of the actor, and the function expressing the priors are complex and non-differentiable. Hence, we assume that the critic is essential to our method.
> 2. We have chosen to report the symmetric best dice score SBD, which was introduced in the popular CVPPP challenge (https://www.sciencedirect.com/science/article/pii/S0167865515003645), as the main metric, as we found that it best represented the segmentation quality. In addition, we report merge and split components of the variation of information to determine whether over- or under-segmentation errors dominate. We decided against using intersection over union based scores, as these yield low scores for the problem at hand due to the issue of matching to objects at a hard threshold for the elongated cells. Nevertheless, our method performs significantly better than the baselines also for IOU based scores. For example, “ours” from Table 1 has scores 0.26, 0.014 and 0.07 for IOU precision at thresholds 0.5, 0.75 and mean average precision, whereas the best baseline “edge + mc” has scores 0.06, 0.002 and 0.01. We can also include these scores in Table 1 if it helps for clarification.
> 3. It would indeed be very interesting to see how far one can go unsupervised compared to fully supervised. We don’t have enough ground-truth in our microscopy dataset to make a fair comparison, but following also other reviewers’ suggestions we have started the experiment with nuclei segmentation. We use convexity-based rewards, and, once the training is finished, will compare to using convexity directly in the network training (the popular StarDist algorithm).

---

> > ### Comment · Reviewer_mhie · 2021-09-04
> > **Great job, but benchmarking is still a problem**
> >
> > I agree with other reviewers that experiments with other datasets are necessary, especially considering that there is a lot of work already published in the specific field of cell segmentation. As I mentioned, it is not necessary to obtain performance above supervised learning, but a comparison of where the method stands would be scientifically useful. I am not changing my rating.

---

### Official Review · Reviewer_dc1u · 2021-07-17

**Rating:** 5
**Confidence:** 4

**Summary:**

In this paper, the authors proposed a reinforcement learning (RL) framework for instance segmentation with superpixels. The proposed framework is based on a stateless actor-critic setup and is demonstrated to be effective on the instance segmentation tasks for one toy dataset and one microscopy dataset.

**Limitations And Societal Impact:**

1) The major concern of this work is the lack of experimental results on public benchmark datasets. For the first toy dataset, the background content seems to be fixed (black content and white ‘noise’). For the second microscopy dataset, the morphological structure of the instances is always special: all the instances overlapping together to form a circle-like fruitfly embryo. To this end, the overall experiments on the two datasets are not sufficient to demonstrate the proposed method can be effective on other general datasets, such as the MS COCO. This made the overall impact of the proposed method questionable.

2) It seems that there are only 10 testing images in the real microscopy dataset, which is a small number. Therefore, a statistical significance test (based on p-value) is suggested for the results on this dataset.

3) Computational complexity analysis is also suggested on the proposed framework.


**Main Review:**

+: strength
-: weakness

1)Originality:

+: This paper proposes to solve the instance segmentation tasks with weak supervision (superpixels) using the RL based methods. Different from the typical weakly supervised image instance segmentation methods, the RL-based one is a new and interesting application.

-: The authors only talked about the related work on RL however ignore the related work on instance segmentation (fully supervised and weakly supervised).

2) Quality:

+: The proposed modules are proved to be effective via the ablation studies on the microscopy dataset. In addition, the authors also provide sufficient analysis on the selection of the hyper-parameters, such as the number of feature channels, random seed, etc.

-: There lack extensive experiments on public instance segmentation benchmark datasets, such as MS COCO [a]. In addition, there are also many instance segmentation tasks in microscopy image analysis, such as nuclei instance segmentation in histopathology images [b], cell instance segmentation in electron microscopy images [1].

[a] https://cocodataset.org/#home
[b] A Multi-Organ Nucleus Segmentation Challenge, in IEEE TMI
[1]: the reference #1 in the manuscript.

3) Clarity:

+: The methods and experiment sections are clearly presented.

-: This paper aims at solving the weakly supervised instance segmentation via RL architecture, however, there lack illustrations and discussion on any related instance segmentation works.

4) Significance:

-: Although the authors have presented many experimental results, there still lack experimental results on representative benchmark instance segmentation datasets on general and medical images, which makes me feel that the overall paper lacks significance.


**Time Spent Reviewing:**

3 hours

---

> ### Author Response · Authors · 2021-08-10
> **Related Work on Instance Segmentation; Additional Experiments**
>
> We would like to thank the reviewer for the summary and the improvement suggestions.
>
> **Related Work:**
>
> We acknowledge that we are missing some discussions of the related work on weakly and fully supervised instance segmentation, including its applications in microscopy. We will add the following paragraph, including new citations, to the Related Work section:
>
> Instance segmentation methods can be divided into proposal-based approaches, which predict the segmentation based on object detections, and proposal-free approaches, which directly group pixels into instances. Proposal-based approaches such as Mask RCNN [1] or DETR [2] are very popular for natural images. Proposal-free approaches encompass different methods such as object boundary prediction followed by agglomeration [3,4,5], pixel embedding prediction followed by clustering [6,7,8] or centroid prediction [9,10].
> Biomedical images are an important application of instance segmentation. Here, proposal-based methods are not commonly used, because they are difficult to extend to volumetric data and yield inferior results for tree-structured or elongated objects [11]. Hence, proposal-free approaches are popular, either based on boundary predictions and agglomeration on pixels [12] or super-pixels [13,14,15] or based on embeddings [16,17].
> Different flavors of weak supervision exist in the context of instance segmentation, for example learning only from bounding box annotations [18,19,20], point annotations [21,22] or sparse object masks [23]. In contrast, our proposed method can learn from pre-defined rules alone, introducing a new category of weak supervision for instance segmentation.
>
> [1] https://arxiv.org/abs/1703.06870
> [2] https://arxiv.org/abs/2005.12872
> [3]https://openaccess.thecvf.com/content_ICCV_2019/html/Gao_SSAP_Single-Shot_Instance_Segmentation_With_Affinity_Pyramid_ICCV_2019_paper.html
> [4]https://openaccess.thecvf.com/content_ECCV_2018/html/Yiding_Liu_Affinity_Derivation_and_ECCV_2018_paper.html
> [5]https://openaccess.thecvf.com/content_iccv_2015/html/Xie_Holistically-Nested_Edge_Detection_ICCV_2015_paper.html
> [6]https://proceedings.neurips.cc/paper/2017/hash/8edd72158ccd2a879f79cb2538568fdc-Abstract.html
> [7]https://arxiv.org/abs/1708.02551
> [8]https://arxiv.org/abs/1904.05257
> [9]https://openaccess.thecvf.com/content_CVPR_2020/html/Cheng_Panoptic-DeepLab_A_Simple_Strong_and_Fast_Baseline_for_Bottom-Up_Panoptic_CVPR_2020_paper.html
> [10]https://openaccess.thecvf.com/content_CVPR_2019/html/Neven_Instance_Segmentation_by_Jointly_Optimizing_Spatial_Embeddings_and_Clustering_Bandwidth_CVPR_2019_paper.html
> [11]https://www.biorxiv.org/content/10.1101/2021.06.09.447748v1.full
> [12]https://ieeexplore.ieee.org/abstract/document/9036993/
> [13]https://www.nature.com/articles/nmeth.4151/
> [14]https://ieeexplore.ieee.org/abstract/document/8364622
> [15]https://elifesciences.org/articles/57613
> [16]https://link.springer.com/chapter/10.1007/978-3-030-00934-2_30
> [17]https://www.nature.com/articles/s41592-020-01018-x
> [18]https://openaccess.thecvf.com/content_cvpr_2017/html/Khoreva_Simple_Does_It_CVPR_2017_paper.html
> [19]https://openaccess.thecvf.com/content_ECCV_2018/html/Anurag_Arnab_Weakly-_and_Semi-Supervised_ECCV_2018_paper.html
> [20]https://link.springer.com/chapter/10.1007/978-3-030-32239-7_72
> [21]https://ieeexplore.ieee.org/abstract/document/9190782
> [22]https://openaccess.thecvf.com/content_ICCV_2019/html/Sofiiuk_AdaptIS_Adaptive_Instance_Selection_Network_ICCV_2019_paper.html
> [23]https://arxiv.org/abs/2103.14572
>
> **Public benchmark experiments:**
>
>  In choosing the datasets for our experiments, we tried to cover two most common problems in microscopy: segmentation of “blob”-like structures and boundary-based segmentation. The reviewer is correct in pointing out that our microscopy dataset has special morphology. As our approach is rule-based, it’s best suited for data where the morphology of the objects can be summarized by rules and not MS COCO-like data which comprises the full complexity of the natural world. This is often the case in biology, where a lot of prior knowledge on object shape and its development has already been accumulated. However, we agree that a public benchmark comparison would improve the paper. We have started the experiments on a standard nuclei segmentation task, using convexity-based rewards. The training is not yet finished, we will report on it here as soon as the results become available.
>
> **Other Limitations:**
>
> - Our metrics are object-based and the 10 images we use contain 983 objects, so the improvement is significant.
> - We don’t think a theoretical analysis of computational complexity is required, as the computational complexity of the components of our method, such as graph neural network or graph partitioning, is known. Empirically, we observe that partitioning (ca. 0.72 seconds per image) is taking the majority of the runtime compared to the network prediction (ca. 0.15 seconds per image).

---

### Official Review · Reviewer_xg8s · 2021-07-20

**Rating:** 6
**Confidence:** 4

**Summary:**

This paper presents a RL approach for instance segmentation which formulates the instance segmentation problem as graph partitioning and the actor critic predicts the edge weights driven by the rewards, which are based on the conformity of segmented instances to high-level priors on object shape, position and size.

**Main Review:**

This paper presents a novel idea of applying RL methodology to instance segmentation. The stateless actor-critic reinforcement learning theme enables non-differential rewards which could incorporate prior knowledge to guide the segmentation, and relaxes the heavy requirement of pixel-wise labeling. The idea is unique and quite interesting.
The related work looks sufficient to me. The experiments are well designed with convincing results.
The writing is overall clear, but there are a few parts which could be improved:
1. For Section3, it's not clear to me the essential difference between Method 1 & 2. It'd be nice to include a graph as Figure 1 to demonstrate Method 2, instead of only including it in Appendix.
2. There is a lot of valuable information in Appendix. It'd be nice to figure out a way to shorten some parts of Method section (such as Section 3.2 and experiments) and include some content from Appendix here.
3. For the experiments, some failure cases and analysis are necessary. It'd be good to add a section of ablation studies.


**Time Spent Reviewing:**

2 hours

---

> ### Author Response · Authors · 2021-08-10
> **Reorganisation of the paper; expansion of alternative approaches; addition of failure cases**
>
> We would like to thank the reviewer for their accurate summary and assessment of our contribution. The comments on how to improve the manuscript are very helpful, we will address them as follows:
> 1. We will merge Fig. 6 (supplementary) into Fig. 1 to present both methods side by side. In addition, we will rename Method 1 to “End-to-end” (E2E Method) and Method 2 to “Pretrained Features” (PF Method) to make it easier to distinguish them throughout the text.
> 2. We agree and will change the text as follows: we will shorten section 3.2, remove duplicates between sections 3.1 and the introductory part of the Methods section and use the freed space to move Appendix A1 into the main text. This way, the description of the E2E Method and the PF method will be in the same section and will hopefully make the whole setup more clear. If the reviewer would like to see some other parts of the Appendix in the main text, we are happy to consider another change, but for now we will follow the general principle of describing the approaches that worked in the main text and the ones that didn’t meet expectations in the Appendix (see also next point).
> 3. For the ablation study, in Table 1 we already compare several configurations of our method. For most of our design choices, we have tried alternative methods, but in most cases these did not yield satisfactory results. Some of these are already described in the Appendix, we will add more about the ones listed below and additionally will make sure they are properly referenced from the main text in order to create a more complete picture of the importance of individual steps.
> Alternative approaches:
>     * Work directly on pixels instead of superpixels: didn’t work as the features were not sufficiently expressive.
>     * Use global Dice score or Focal loss instead of the subgraph Dice score: it works, but not as well as the subgraphs
>     * Instead of the final FC layer in the GNN use graph pooling down to a single feature vector: it works, but not as well as the final setup.
>     * Instead of Multicut for graph partitioning, use agglomerative graph clustering and k-means with node-features and multistep RL where we predicted movement vectors for node feature states
>     * Instead of SAC use A2C [Mnih et al, ICML 2016] and ACER [Wang et al, ICLR 2017]. Both were very sensitive to local optima. Direct Q-learning led to exploding gradients. The (really small) momentum of the 2 Q-functions in SAC is of huge help in our setup.
>     * Use sphere embeddings and cosine distance as well as with "free" embeddings and l2 (distance used for kmeans and agglomerative clustering), not nearly as good as the final setup.
>
> We also agree that a description of failure cases would improve the paper, we will add our general observations to the Discussion and show qualitative examples in the Appendix. Briefly, the failure cases tend to cluster in the areas where the rewards are imprecise. As an example, note the leftmost part of the images in Fig. 5, where the cells are more curved than in the other parts of the embryo. Additionally, the background sometimes gets over-split. While this problem is not of big practical concern as large background pieces can easily be merged in post-processing, we believe that fixing them directly in our setup would need better rewards for the background areas, including a better incorporation of semantics. Similarly, when the boundary evidence is weak, cells can get merged into the background, this case is not penalized enough by our current rewards.
>
> Limitations and Societal Impact: see comments for Reviewer 1 (Reviewer iAQ6).

---

> > ### Comment · Reviewer_xg8s · 2021-08-12
> > **Response to the authors' rebuttal**
> >
> > Thank you for the detailed explanation and modification plan. Hope these modifications will make the paper more impactful and easier to follow and reproduce the results.

---

### Official Review · Reviewer_iAQ6 · 2021-07-20

**Rating:** 4
**Confidence:** 3

**Summary:**

The paper proposes the use of RL for instance segmentation. In particular, the paper uses super pixels together with graph partitioning and RL to learn the labels of instances. The model is validated on two datasets, one synthetic and one biomedical imaging dataset and is numerically compared to two approaches ([28] and [47]).

**Limitations And Societal Impact:**

There is no limitation section/paragraph nor societal impact section/paragraph. Please add those to the conclusions section.

**Main Review:**

**Originality:**

In the paper, the main motivation of the use of RL is to enable optimization of discrete objective. However, this use of RL is standard. Moreover, given this paper motivations, the reviewer would expect to see empirical comparisons to alternative methods to learn discrete assignments  (e.g. straight through, DETR-style https://arxiv.org/abs/2005.12872, [8]). Thus, from the methodological point of view the novelty is rather limited. From the application point of view, the instance segmentation of light microscopy images, the paper might have some edge of originality; however, it is hard to assess it given the niche domain. To fully appreciate the originality of the method it would be beneficial to validate the approach on a more standard benchmark.


**Clarity:**

The paper is generally clearly written. However, the clarity of the methods could be improved. The presentation of this section is a bit convoluted making it hard to grasp the overall picture of the proposed approach. The reviewer would encourage the authors to simplify the presentation of the Methods, especially focusing on the aspects of why particular decisions were made. E.g. why are the authors considering method 1 and 2? Why stateless RL with sequence length of 1? Why not use straight through?

For a non-microscopy expert, it is hard to understand the color codings of Figs 4 and 5.


**Quality and significance:**

The quality and significance of the results are hard to assess given the niche application domain. It would be beneficial to test the ideas on some more standard benchmarks to instance image segmentation (e.g. COCO).

Additional questions comments:
- Why are superpixels a good prior for this task?
- Would it be possible to use other priors/partitioning methods?
- It would be nice to discuss the tradeoffs between added computational costs due to the use of RL and metric gains.
- Would it be possible to report the numbers of a standard instance segmentation model for this task, (e.g. https://arxiv.org/abs/1703.06870 or some followup)?


**Time Spent Reviewing:**

3-4

---

> ### Author Response · Authors · 2021-08-10
> **Originality of the Proposed Method; Clarity; Additional Experiments; Limitations**
>
> It appears that we have not managed to fully highlight the originality of our work. It is true that (stateful) RL has been used to optimize a discrete objective. Unlike this prior work, we use much simpler stateless RL and show that this formulation is very well suited to solve an instance segmentation problem without fully labelled groundtruth. Our key contribution is the replacement of pixelwise ground-truth by the prior knowledge of the segmented object appearance - that we refer to as “rules'' or “priors”. Our formulation enables us to use a wide array of rules without restrictions on them being differentiable (as is the case for end-to-end DL methods) or separable (as is the case for probabilistic graphical models). To the best of our knowledge - supported also by the opinions of other reviewers - nothing similar has been proposed before. Related prior work [8] and [9] make use of other RL techniques in the context of instance segmentation, see lines 59-67. However, both these methods require fully annotated ground-truth labels and are stateful. Similarly, DETR that was pointed out by the reviewer requires fully labeled ground-truth for training with the bipartite matching loss, so we can’t directly compare to it. Straight-through estimators STE (https://arxiv.org/pdf/1308.3432.pdf) are also not applicable: in order to compute the loss, we need to transform the edge weights into an instance segmentation using graph partitioning and score the resulting instance segmentation using pre-defined rules. Both steps involve complex and non-differentiable functions, going well beyond the capability of STE to provide meaningful gradients, which is typically applied to hard nonlinearities such as thresholding or stochastic neurons. We will, however, mention STE as  another approach to overcome non-differentiable steps in training in the Related Work.
>
> With this, we hope to have clarified our main contribution and shown that its originality is not limited to what the reviewer refers to as a “niche domain”. Furthermore, we don’t consider the instance segmentation for microscopy images to be “niche”. It has broad applications in medicine and life sciences and is a well studied topic, regularly addressed at NeurIPS and other top ML/CV conferences. In order to highlight this point, we will add a new paragraph on instance segmentation and its applications in microscopy to the related work (see response to Reviewer 3 (dc1u) for the full text with references).
>
> We agree with the reviewer’s comments on the clarity of the paper and will make the setup description more concise. In general, we aimed to state our choices in the main paper and present the weaker alternatives in the Appendix. Following the reviewer’s comment, we will make sure that all alternatives are properly described in the Appendix and especially that they are referenced from the main paper. Following also other reviewers’ comments, we will merge Figure 1 and Figure 6 of the Appendix, move the Appendix section A1 to the main text for easier comparison of the two proposed methods and rename the methods themselves to “E2E Method” for end-to-end, previously Method 1, and “PF Method” for “precomputed features”, previously Method 2.
>
> Concerning evaluation on MS COCO, we do not consider our method to be the best fit for this dataset. As our approach is rule-based, we would have to come up with strong  priors for all the classes found in MS COCO which is nearly impossible considering the variability of the natural world. We are currently working on a semi-supervised extension of our approach and hope that it can address MS COCO-like tasks better than the unsupervised segmentation presented in this work, best suited for segmentation of regular objects in difficult conditions of high, potentially structured, noise and blur. However, we appreciate the point of extending the experiments section and, following also the suggestions of other reviewers, we will introduce another microscopy dataset which is a more standard benchmark for this kind of method. We have started the experiments, but they are not yet finished training, we will report on them here as soon as the results become available.
>
> We appreciate the comment about the color codings in Figs. 4 and 5. We noticed that the color coding in the groundtruth panels of Fig. 4 is confusing and will correct them both to have the same random color table. We’d be very grateful to have other pointers to problems that  make these figures unclear. We are ready to improve, but we are not certain about the way, as PCA for embeddings and random colors for instance segmentations are a common way of visualisation.
>
> Additional points raised by the reviewer:
> - “Why are superpixels a good prior for this task?” Superpixels are routinely used in microscopy image segmentation and especially in the methods which rely on graph partitioning because these are often computationally expensive (see the new paragraph in Related Work, suggested above). Besides making the problem smaller, superpixels allow to compute more expressive features than pixels, which we either use as fixed (“PF Method”, previously Method 2) or further improve in training (“E2E Method”, previously Method 1). To avoid confusion, we would like to point out that we use the term prior very differently in the manuscript and use it to refer to the rule-based learning objectives.
> - “Would it be possible to use other priors/partitioning methods?” Yes, any prior or partitioning method can be used.
> - “It would be nice to discuss the tradeoffs between added computational costs due to the use of RL and metric gains.” The overhead of network prediction (ca. 0.15 seconds per image) is negligible compared to graph partitioning (ca. 0.72 seconds per image). Note that our method achieves significantly better scores than the baselines, cf. Table 1, including “edge + mc”, which also uses graph partitioning, but with edge weights derived directly from the predictions of the pretrained boundary network.
> - “Would it be possible to report the numbers of a standard instance segmentation model for this task, e.g. Mask RCNN?” Mask RCNN and other proposal-based methods are not particularly effective on microscopy data, presumably because of the lack of an ImageNet-like dataset to train an appropriate backbone, see for example https://www.biorxiv.org/content/10.1101/2021.06.09.447748v1.full. For most standard supervised approaches, we do not have enough labeled data for a fair comparison. We do compare with embeddings-based methods (ref. 28 and Table 1), which are normally used for natural images and can be considered fairly standard.
>
> Concerning Societal Impact, we followed the guidelines at https://neurips.cc/Conferences/2021/PaperInformation/PaperChecklist and assumed that we only need to discuss the negative societal impact, which we do not expect at all from our work. We are happy to comment on the positive sides more explicitly in the Discussion. In particular we envision that our work will lead to the overall reduction of manual annotation time for many biomedical segmentation tasks, freeing the domain experts to carry out their core research instead of pixelwise labelling the images.
>
> Finally, we agree that Limitations should be summarized in the Discussion section, for which the 2nd paragraph will now read as follows:
> “We performed proof-of-concept experiments to demonstrate the feasibility of our approach on synthetic and real data and showed in particular that our setup can segment microscopy images with no direct supervision other than high-level reasoning. While the absence of pixelwise supervision brings a lot of obvious benefits such as the manyfold reduction of tedious annotation effort, it introduces some limitations to the applicability of the approach. It requires strong priors to generate good rewards and is therefore best suited for datasets where the expected objects’ shape, position or size can be summarized in a few rules. In the future, we plan to fully explore the semi-supervised setup (briefly introduced in section A.11) where we believe these limitations can be alleviated. Furthermore, even in case of full supervision with ample groundtruth, our RL-based formulation enables end-to-end instance segmentation with direct object-level reasoning, which will allow for post-processing-aware training of the CNN which predicts object boundaries or embeddings.”

---

### Author Response · Authors · 2021-09-04
**Preliminary experiments with nuclear segmentation**

We apologize for the delay in getting the new results, it took us a while to adjust to an intensity-based rather than boundary-based setting.

We have conducted initial experiments on the data from the Data Science Bowl Nucleus Segmentation challenge, the reference dataset for nucleus segmentation in biomedical imaging. For this experiment, we have used the following reward. The edges actions are penalized by comparison of the mean absolute difference in raw pixel intensities belonging to the two superpixels connected to that edge. It is clear that this reward is quite noisy and that we might as well just take directly the absolute difference in mean pixel intensities as the weights for the Multicut. Therefore we inject additional prior knowledge about the Data by inducing size constraints. E.g if we evaluate an edge which is connected to two superpixels where we are confident that those are background superpixels judging from their size, we penalize actions representing unmerge affinities. Same holds by evaluating a superpixel which is likely to be foreground and which is connected to a superpixel which is likely to be background we penalize actions representing merge affinities.

With this set-up, we can achieve segmentation results that are comparable to a similar approach without supervision, which uses the same superpixels and graph and estimates weights for the graph partition problems from an edge image filter. While not competitive with supervised approaches yet, we see visually that the node embeddings learned are clearly semantically split into foreground and backgound and we observe that the network, which was only trained for about a day, is still improving. Consequently we hope that with more training, and potentially an improved reward function, our approach could out-perform the baseline unsupervised approaches and narrow the gap to the supervised approaches, especially in a semi-supervised setting.

---

### Decision · Program_Chairs · 2021-09-27

**Decision:**

Reject

**Comment:**

This paper proposes to use stateless RL for instance segmentation on microscopy datasets. Initially it received mixed reviews, upon the rebuttal and further discussion, reviewers are more aligned that the approach is novel, however they agreed that the benchmark dataset that the authors used were probably too small. The authors provided an update but that was after the discussion deadline so I don't think that can be considered for this round. Hence AC recommends rejection but the authors should consider adding experiments on public benchmark datasets and submit to the next available venue.